

# Automated essay scoring with SBERT embeddings and LSTM-Attention networks

Yuzhe Nie

School of Foreign Languages, Shanghai University, Shanghai, China

## ABSTRACT

Automated essay scoring (AES) is essential in the field of educational technology, providing rapid and accurate evaluations of student writing. This study presents an innovative AES method that integrates Sentence-BERT (SBERT) with Long Short-Term Memory (LSTM) networks and attention mechanisms to improve the scoring process. SBERT generates embedding vectors for each essay, which are subsequently analyzed using a bidirectional LSTM (BiLSTM) to learn the features of these embedding vectors. An attention layer is introduced to enable the system to prioritize the most significant components of the essay. Evaluated using a benchmark dataset, our approach shows significant improvements in scoring accuracy, highlighting its ability to improve the reliability and efficiency of automated assessment systems.

## INTRODUCTION

Automated essay scoring (AES) systems utilize computer algorithms to assess and grade essays through the analysis of their textual content. The systems are typically comprised of two primary components: a feature extraction module that collects linguistic data, including word count, grammar, and syntactic complexity, and a scoring module that evaluates and assigns grades based on these features. AES models demonstrate an ability to provide scores that frequently match closely with human evaluations. The development of AES can be traced back to 1966, when *Page (1966)* introduced the Project Essay Grader (PEG). This innovative statistical method established a connection between the surface characteristics of writing—like word length and sentence complexity—and the scores assigned by human evaluators. Initial AES systems mainly depended on features and scoring criteria that were manually developed to replicate human evaluation. *Palmer, Williams & Dreher (2002)* presented the application of Latent Semantic Analysis (LSA) for evaluating essay content through the measurement of semantic similarity among words in a text, thereby enhancing the content evaluation dimension of AES. Despite the advancements made, *Reilly (2013)* highlighted concerns regarding possible biases in AES, especially within the context of Massive Open Online Courses (MOOCs), where differences between machine and human grading were noted. Later developments in AES involved the use of regression models by *Alikaniotis, Yannakoudakis & Rei (2016)* to evaluate essays by analyzing linguistic features linked to essay quality. *Taghipour & Ng*

Corresponding author
Yuzhe Nie, nieyuzhe1900@163.com

*(2016)* illustrated how deep learning models, particularly neural networks, can more effectively capture the complexities of essay quality. In a recent study, *Cozma, Butnaru & Ionescu (2018)* investigated the combination of character n-grams and word embeddings, demonstrating enhanced performance relative to previous methods.

With the rise of neural network architectures and natural language processing methods (*Wu et al., 2023*; *Gu et al., 2023*; *Ding et al., 2023*), the progress in data accessibility and computational frameworks has driven the development of AES (*Li & Jianxing, 2024*). Consequently, there has been increasing emphasis on enhancing scoring criteria to incorporate various aspects of writing quality, resulting in more comprehensive evaluations (*Carlile et al., 2018*). Among the innovative techniques, Bidirectional Encoder Representations from Transformers (BERT) has emerged as a powerful tool or AES. The capacity to recognize hidden contextual relationships within text has demonstrated superior performance compared to conventional models. For example, *Wang et al. (2022)* demonstrated the advantages of BERT in learning multi-scale essay representations, resulting in enhanced performance compared to models based on Long Short-Term Memory (LSTM).

LSTM networks, known for their effectiveness with sequential data, have gained significant popularity in AES applications. *Janda (2019)* highlighted the effectiveness of LSTM models in monitoring semantic shifts across an essay, demonstrating their capability to capture gradual changes in meaning. Additionally, *Attali & Burstein (2004)* investigated the problem of essay length bias in automated essay scoring and discovered that LSTM-based models might mitigate these biases, leading to more reliable and equitable evaluations. Another innovative direction in AES exploration involves the integration of coherence features into scoring models. *Farag, Yannakoudakis & Briscoe (2018)* emphasized that modeling the logical flow of ideas can improve the accuracy of essay evaluation by concentrating on the structural integrity of the text. This perspective is further supported by *Uto, Xie & Ueno (2020)*, who proposed that the integration of handcrafted features with neural network methodologies could enhance scoring accuracy.

Alongside neural networks, ensemble methods have become increasingly prominent in AES studies due to their capacity to combine various scoring features. *Nadeem et al. (2019)* introduced neural models that are sensitive to discourse, integrating various essay features to enhance scoring precision, which aligns with the increasing focus in automated essay scoring on multidimensional evaluation. This transition indicates a shift from one-dimensional scoring and moves towards a more comprehensive assessment, taking into account elements like coherence, argumentation, and content quality in addition to linguistic characteristics (*Carlile et al., 2018*). These advancements represent an important advancement in improving the complexity and reliability of AES systems.

## MATERIALS AND METHODS

### Data collection and preprocessing

For this study, we utilized the Automated Student Assessment Prize (ASAP) dataset (*Hamner et al., 2012*), a widely recognized benchmark for evaluating AES systems. This dataset was originally introduced as part of a shared task designed to compare AES system

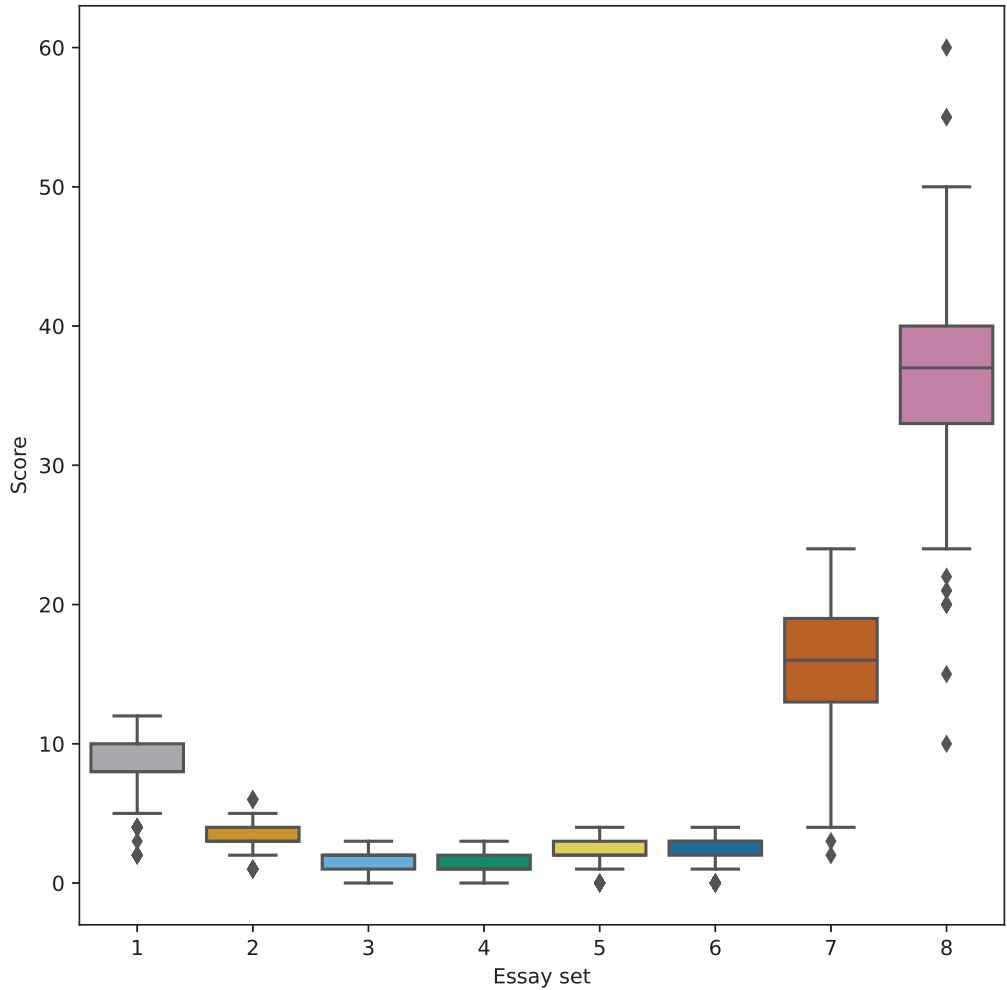

**Figure 1 Score distribution of each essay set.**

performance against human-assigned scores. The essays in the ASAP dataset were written by middle school students in the U.S., ranging from grades 7 to 10. The dataset consists of essays responding to eight distinct prompts, each characterized by unique language features such as varying levels of concreteness, open-endedness, and scoring scales (as outlined in Fig. 1). An overview of the dataset's structure is provided in Table 1.

To prepare the textual data for analysis, we first cleaned the raw text by removing non-alphabetic characters, except for certain punctuation marks necessary for preserving the integrity of sentence structure. Following this, the text was tokenized into individual words, and English stopwords were filtered out to reduce noise in the data. We then applied stemming to each token, transforming words to their root forms, which helps to standardize variations of words for better analysis in natural language processing tasks. The result was a preprocessed text string, optimized for further AES model development.

We divided the dataset into three subsets: training, validation, and test sets, in a 70:10:20 ratio. This resulted in 9,342 essays for training, 1,038 for validation, and 2,596 for testing.

**Table 1 Statistics of the ASAP dataset.**

| Prompt | Essays | Avg length | Score range | WordPiece length |
|---|---|---|---|---|
| 1 | 1,783 | 350 | 2–12 | 649 |
| 2 | 1,800 | 350 | 1–6 | 704 |
| 3 | 1,726 | 150 | 0–3 | 219 |
| 4 | 1,772 | 150 | 0–3 | 203 |
| 5 | 1,805 | 150 | 0–4 | 258 |
| 6 | 1,800 | 150 | 0–4 | 289 |
| 7 | 1,569 | 250 | 0–30 | 371 |
| 8 | 723 | 650 | 0–60 | 1,077 |

This division ensured that the model had sufficient data to learn, validate its performance, and finally be tested on unseen essays for unbiased evaluation.

## Rationale for model selection

Many deep learning architectures employed for essay scoring are relatively simplistic and do not fully leverage the unique features embedded in the data. To address this limitation, we designed a computational framework based on Sentence-BERT (SBERT) (*Reimers & Gurevych, 2019*) and LSTM with attention mechanisms, aimed at enhancing both the semantic processing and predictive capabilities of the model in natural language tasks.

SBERT, built on the transformer architecture, generates embedding vectors that capture the meaning of individual sentences more effectively than traditional models. Its ability to create high-quality sentence representations allows for better identification and comparison of semantic nuances within essays. Transformer-based models, such as SBERT, are well-known for their attention mechanisms, which have gained prominence for their ability to focus on the most important parts of the input data (*Vaswani et al., 2017*; *Leow, Nguyen & Chua, 2021*; *Nguyen-Vo et al., 2021*; *Badaro, Saeed & Papotti, 2023*). These mechanisms allow the model to selectively attend to key elements in an essay, improving its ability to grasp the deeper semantic relationships.

Furthermore, incorporating LSTM with attention mechanisms adds another layer of sophistication to the model. LSTMs excel at processing sequential data by retaining critical information over time, and when combined with attention, they can focus on the most relevant parts of the sequence. This combination enhances the model's capacity to handle long sequences without losing important context, ultimately improving the accuracy of essay scoring.

By integrating SBERT's powerful sentence embeddings with LSTM's attention-enhanced sequential processing, this model not only optimizes performance in scoring tasks but also provides deeper insights into complex linguistic patterns, making it a robust tool for analyzing and evaluating essays.

## Assessment metrics

To assess the effectiveness of the models, we employed various evaluation metrics including Mean Squared Error (MSE), the coefficient of determination ($R^2$), Mean

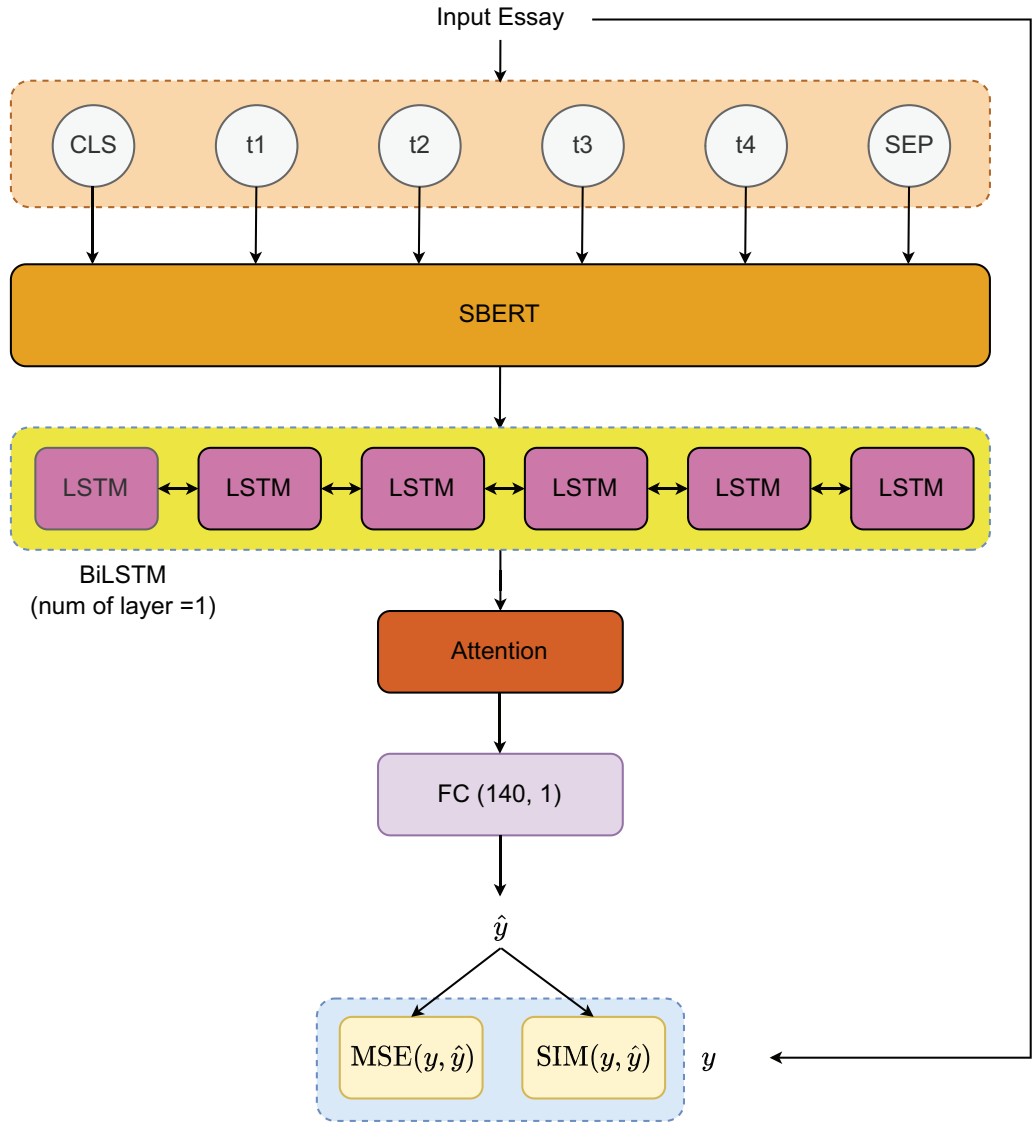

$$\text{Loss}_{\text{total}}(y, \hat{y}) = \alpha \, \text{MSE}(y, \hat{y}) + \gamma \, \text{SIM}(y, \hat{y})$$

**Figure 2  Model architecture.**   

Absolute Error (MAE), Root Mean Square Deviation (RMSE), and Quadratic Weighted Kappa (QWK). This allows us to conduct a comprehensive evaluation of the precision and consistency of the predictions.

## Proposed method

### Model architecture

To develop the essay scoring system, we propose the architecture illustrated in Fig. 2. The process begins with a pre-trained sequence transformer for extracting features from the input essays. These features are then processed through a bidirectional Long Short-Term Memory (BiLSTM) layer, which effectively captures the underlying patterns in the data. To

further refine the model's attention to critical essay components, an attention mechanism is applied to the output from the BiLSTM layer. The final essay scores are generated through a fully connected (FC) layer with dimensions (140, 1). For optimization, we utilize two loss functions—MSE and cosine similarity (SIM)—which will be explained in detail in the "Loss Functions" section.

### Essay representation

To represent essays at both the token and document levels, we utilize a pre-trained BERT model, augmented by a BiLSTM network and an attention mechanism. The process begins by tokenizing each essay using the BERT tokenizer, resulting in a token sequence $T_1 = [t_1, t_2, \ldots, t_n]$, where $t_i$ is the $i$-th token and $n$ represents the total number of tokens. BERT's tokenization follows the WordPiece method, and since BERT's maximum input sequence length is 512 tokens, we construct a new sequence $T_2$ from $T_1$ as follows:

$$
\begin{cases}
[CLS] + [t_1, t_2, \ldots, t_L] + [SEP], & \text{if } n > L \\
[CLS] + T_1 + [SEP], & \text{if } n = L \\
[CLS] + T_1 + [PAD](L - n) + [SEP], & \text{if } n < L.
\end{cases}
\tag{1}
$$

Here, $L = 510$ is the maximum sequence length allowed for tokens between the special tokens $[CLS]$ and $[SEP]$, marking the start and end of the sequence, respectively. If the essay contains fewer tokens than $L$, padding is applied to maintain the fixed sequence length. The token, segmentation, and position embeddings are then combined to create the final input representation fed into BERT.

After obtaining the contextualized token embeddings from BERT, we use a BiLSTM network to capture the sequential dependencies within the essay. The BiLSTM processes the token sequence $H = [h_1, h_2, \ldots, h_L]$, where $h_i$ represents the hidden state corresponding to token $t_i$. The forward and backward passes of the LSTM are defined as:

$$
h_i^{\rightarrow} = \text{LSTM}_{\text{fwd}}(t_i), \quad h_i^{\leftarrow} = \text{LSTM}_{\text{bwd}}(t_i).
\tag{2}
$$

The final output for each token from the BiLSTM is the concatenation of the forward and backward hidden states: $h_i = [h_i^{\rightarrow}, h_i^{\leftarrow}]$.

To further enhance the model's focus on key tokens, we apply a self-attention mechanism over the BiLSTM outputs. The attention score $\alpha_i$ for the $i$-th token is computed as:

$$
\alpha_i = \frac{\exp(W h_i)}{\sum_{j=1}^{L} \exp(W h_j)},
\tag{3}
$$

where $W$ is a learnable weight matrix. The final essay representation $v$ is then obtained as a weighted sum of the BiLSTM outputs:

$$
v = \sum_{i=1}^{L} \alpha_i h_i.
\tag{4}
$$

This representation $v$ serves as the input for downstream tasks, such as classification or regression, depending on the application.

## Loss functions

MSE is a metric used to measure the average squared differences between the predicted scores and the actual labels, defined as follows:

$$\text{MSE}(y, \hat{y}) = \frac{1}{N} \sum_{i=1}^{N} (y_i - \hat{y}_i)^2, \tag{5}$$

where $y_i$ represents the true score of the $i$-th essay, $\hat{y}_i$ is the corresponding predicted score, and $N$ is the total number of essays evaluated. MSE penalizes larger errors more heavily, which can improve the overall accuracy of the model in cases where large errors are unacceptable. Because the essays have a wide range of scores, this loss function will help enhance the model's predictions.

To assess the similarity between two vectors, we use the SIM function, which measures the alignment between vectors based on their orientation. The SIM loss helps assess the similarity of the input texts since there will be many texts with high similarity in the data. This allows the model to produce more accurate and clearer results for essays that are highly similar. During training, the SIM loss encourages the model to recognize similar pairs of vectors, enhancing its ability to capture relationships within the batch of essays. The SIM loss is defined as:

$$\text{SIM}(y, \hat{y}) = 1 - \cos(y, \hat{y}). \tag{6}$$

The total loss function combines these two components, MSE and SIM, into a single objective, formulated as:

$$\text{Loss}_{\text{total}}(y, \hat{y}) = \alpha \text{MSE}(y, \hat{y}) + \beta \text{SIM}(y, \hat{y}), \tag{7}$$

where $\alpha$ and $\beta$ are weight parameters optimized based on the model's performance on the validation set.

# EXPERIMENTAL RESULTS AND DISCUSSION

We trained our model for 50 epochs with a learning rate of 0.01, utilizing the Adam optimizer (*Kingma & Ba, 2014*). The training process was implemented using PyTorch 2.0.0 and executed on an RTX 3060 GPU with 12 GB of memory. All computations were conducted on a machine running Windows 11, equipped with an AMD Ryzen 7 5800X 8-Core Processor (3.80 GHz) and 32 GB of RAM.

## Baseline and benchmarking models

To evaluate the effectiveness of the LSTM-Attention mechanism in our model, we established a variety of baseline methods using SBERT embeddings. These baseline models include widely-used machine learning techniques, such as BERT combined with Support Vector Machines (SVM (*Pal & Mather, 2005*)), Random Forests (RF (*Breiman, 2001*)), $k$-Nearest Neighbors (KNN (*Kramer, 2013*)), Extreme Gradient Boosting (XGB (*Chen &*

**Table 2 Comparison results with baseline models.**

| Models | MSE | $R^2$ | MAE | RMSE | QWK |
|---|---|---|---|---|---|
| LSTM-based Model | 18.3864 | 0.6315 | 2.7686 | 4.2879 | 0.4748 |
| LSTM-Attention | 13.0475 | 0.7643 | 2.3010 | 3.6121 | 0.5862 |
| BERT + Random Forest | 6.1241 | 0.8842 | 1.5280 | 2.4747 | 0.6987 |
| BERT + Support Vector Machines | 13.0475 | 0.7643 | 2.3010 | 3.6121 | 0.5862 |
| BERT + $k$-Nearest Neighbor | 11.6149 | 0.7965 | 2.1703 | 3.4081 | 0.6132 |
| BERT + $e$Xtreme Gradient Boosting | 10.6173 | 0.8167 | 2.0743 | 3.2584 | 0.6426 |
| BERT + LSTM | 7.0722 | 0.8876 | 1.6766 | 2.6594 | 0.7302 |
| BERT + CNN | 6.0048 | 0.8859 | 1.5268 | 2.4505 | 0.7125 |
| Ours | **4.7645** | **0.9286** | **1.3544** | **2.1828** | **0.7876** |

Note:
Bold indicates the best performance.

Guestrin, 2016)), and 1D Convolutional Neural Networks (CNN1D (Kiranyaz et al., 2021)). Additionally, we incorporated a BERT + LSTM configuration to specifically assess the contribution of the attention mechanism. These traditional LSTM and LSTM-Attention models serve as foundational baselines for comparison, particularly in sequence-based learning tasks.

For a comprehensive performance analysis, we benchmarked our proposed model against several well-established AES approaches. These include Tran-BERT-MS-ML-R (Wang et al., 2022), which utilizes BERT's robust contextual embeddings to capture intricate essay features, and XLNet (Jeon & Strube, 2021), which addresses the challenge of essay length variability and its effect on scoring accuracy. We also compared our model to SkipFlow (Tay et al., 2018), a system that emphasizes coherence as a key factor in the scoring process, and Many Hands Make Light Work (MHMLW (Kumar et al., 2021)), which focuses on essay-specific traits to enhance assessment reliability. Furthermore, we evaluated against Automatic Features (AF (Dong & Zhang, 2016)), which provides a broad analysis of feature extraction techniques, and Flexible Domain Adaptation (FDA (Phandi, Chai & Ng, 2015)), known for its innovative methods in adapting scoring models across multiple domains.

By systematically comparing our model to these approaches, we aim to demonstrate the efficiency of our architecture and its potential advantages in improving automated essay scoring, particularly in terms of semantic understanding, feature extraction, and handling complex essay characteristics.

## Performance comparison with baseline models

The performance comparison between our proposed model and several baseline models is illustrated in Table 2. The model exhibited exceptional performance across various evaluation metrics, including MSE, $R^2$, MAE, RMSE, and QWK. It is important to highlight that it achieved the lowest MSE of 4.7645 and an RMSE of 2.1828, demonstrating a significant decrease in prediction error relative to other models. Additionally, our model achieved the highest $R^2$ value of 0.9286, indicating a robust correlation between the

**Table 3 Comparison results with other benchmarking models.**

| Models | MSE | $R^2$ | MAE | RMSE | QWK |
|---|---|---|---|---|---|
| AF (*Dong & Zhang, 2016*) | 20.8331 | 0.5604 | 2.9501 | 4.5643 | 0.4448 |
| FDA (*Phandi, Chai & Ng, 2015*) | 16.4041 | 0.6840 | 2.6094 | 4.0502 | 0.5157 |
| MHMLW (*Kumar et al., 2021*) | 10.0127 | 0.8304 | 2.0055 | 3.1643 | 0.6532 |
| Tran-BERT-MS-ML-R (*Wang et al., 2022*) | 9.0755 | 0.8490 | 1.9112 | 3.0126 | 0.6891 |
| XLNet (*Jeon & Strube, 2021*) | 8.3130 | 0.8640 | 1.8147 | 2.8832 | 0.7042 |
| SkipFlow (*Tay et al., 2018*) | 6.2887 | 0.8804 | 1.5542 | 2.5077 | 0.6820 |
| Ours | **4.7645** | **0.9286** | **1.3544** | **2.1828** | **0.7876** |

Note:
Bold indicates the best performance.

predicted and actual essay scores. Furthermore, the model achieved a QWK of 0.7876, indicating a strong alignment with human-assigned scores, which further confirms its scoring accuracy.

Among the baseline models, the combinations of BERT with machine learning algorithms, including BERT + CNN and BERT + Random Forest, demonstrated competitive performance, especially in reducing prediction errors and improving accuracy. This highlights the importance of utilizing advanced transformer-based embeddings such as BERT for effective feature extraction. Nonetheless, in spite of their strong performance, these models did not reach the accuracy levels of our proposed LSTM-Attention model, highlighting the advantages of attention mechanisms in enhancing prediction precision.

In contrast, sequence-based models like LSTM and LSTM-Attention, when used without embeddings from pre-trained language models, displayed the lowest performance, with MSEs of 18.38 and 13.05, respectively. This underscores the importance of leveraging pre-trained embeddings, such as SBERT, particularly for tasks involving smaller, less diverse datasets. The incorporation of SBERT enabled our model to learn additional semantic and contextual details, resulting in enhanced accuracy in essay scoring results.

## Performance comparison with benchmarking models

The findings illustrated in Table 3 provide additional validation of the efficacy of our proposed model across all performance metrics. Our model demonstrated a remarkable performance with a MSE of 4.7645, showing a substantial decrease in prediction errors when compared to other models. This highlights its accuracy in predicting essay scores. Furthermore, the model achieved an impressive $R^2$ value of 0.9286, accounting for around 92.86% of the variance in the dataset, indicating a strong alignment with the underlying data distribution.

The MAE of 1.3544 indicates the average magnitude of errors, demonstrating the model's capacity to produce predictions with minimal differences. Furthermore, the RMSE of 2.1828 highlights the strength of our methodology, as this metric places greater emphasis on larger errors, and the comparatively low value suggests a high level of prediction accuracy.

| Trial | MSE | $R^2$ | MAE | RMSE | QWK |
|---|---|---|---|---|---|
| 0 | 4.7645 | 0.9286 | 1.3544 | 2.1828 | 0.7876 |
| 1 | 4.8393 | 0.9282 | 1.3602 | 2.1998 | 0.7877 |
| 2 | 4.7540 | 0.9297 | 1.3514 | 2.1804 | 0.7925 |
| 3 | 4.8210 | 0.9270 | 1.3659 | 2.1957 | 0.7897 |
| 4 | 4.7441 | 0.9281 | 1.3487 | 2.1781 | 0.8005 |
| 5 | 4.8291 | 0.9284 | 1.3593 | 2.1975 | 0.7872 |
| 6 | 4.8287 | 0.9279 | 1.3583 | 2.1974 | 0.7853 |
| 7 | 4.7160 | 0.9306 | 1.3462 | 2.1716 | 0.7885 |
| 8 | 4.7829 | 0.9281 | 1.3540 | 2.1870 | 0.7918 |
| 9 | 4.7353 | 0.9302 | 1.3541 | 2.1761 | 0.7922 |
| Mean | 4.7815 | 0.9287 | 1.3553 | 2.1866 | 0.7903 |
| STD | 0.0450 | 0.0011 | 0.0058 | 0.0103 | 0.0043 |

**Table 4 Performance of the proposed model across multiple trials.**

Moreover, the QWK score of 0.7876 highlights the model's ability in capturing and correlating with human-assigned scores, while also considering the severity of differences between predicted and actual scores. This metric highlights the model's capability in evaluating essays with a significant level of consistency and dependability.

In comparison, although models like SkipFlow and XLNet showed notable performance, achieving QWK scores of 0.6820 and 0.7042 respectively, our model surpassed them across all metrics. This demonstrates the enhanced capability of our architecture to utilize key essay characteristics, leading to improved and reliable scoring results.

## Stability analysis

To evaluate the stability of our proposed model, we conducted additional experiments by repeating the data splitting and training process nine more times, each using a different random seed. This approach ensures that the reported performance is not influenced by a specific data split and provides a robust assessment of the model's consistency.

Table 4 summarizes the performance of our model across all 10 trials. Trial 0 corresponds to the results presented in Tables 2 and 3, while trials 1 through 9 represent additional runs with randomized data splits. For each trial, we evaluated the model using the same metrics as in the original experiments, ensuring comparability.

As observed in Table 4, our model demonstrates stable performance across all evaluation metrics, with minimal variance. This consistency highlights the robustness of our model and its ability to generalize well across different dataset splits. The small variance in all the performance metrics further emphasizes the reliability of our approach.

## Limitations and future directions

While our proposed model demonstrates strong performance in AES, several limitations should be acknowledged. First, the dataset used in this study, though widely recognized as

a benchmark, is relatively limited in terms of diversity, as it consists primarily of essays written by middle school students in the U.S. This restricts the generalizability of our findings across different age groups, educational levels, or cultural contexts. Expanding the scope to include a broader range of essays, such as those from higher education or essays written in different languages, could provide more comprehensive insights into the model's robustness.

Second, while the SBERT embeddings proved effective in capturing semantic nuances, further exploration into other advanced transformer-based models, such as GPT or T5, might offer additional performance improvements. Additionally, the current implementation of the LSTM-Attention mechanism, while effective, could be enhanced by experimenting with more sophisticated attention mechanisms, such as multi-head attention or dynamic attention models, to further refine the focus on critical essay components.

Another limitation is the computational intensity required for training transformer-based models, which may present challenges for scalability in real-world applications. Future work could explore optimization techniques or model compression methods to reduce the computational load without sacrificing performance. Furthermore, future research could also investigate the application of this model in adaptive learning systems, where real-time essay scoring and feedback could be integrated into personalized learning platforms.

Future directions for this study could explore leveraging advancements in open-vocabulary models and synthetic data augmentation techniques to enhance the scalability and robustness of AES systems. For instance, integrating approaches similar to those used in *Shi, Dao & Cai (2024)* and *Shi, Hayat & Cai (2024)* could enable AES models to generalize across diverse linguistic and cultural contexts by adapting to new essay topics and vocabularies without extensive retraining. Additionally, addressing the dataset diversity limitations noted in this study, techniques like synthetic data generation inspired by *Wang, Chukova & Nguyen (2023)* could enrich the training dataset with balanced and representative samples from underrepresented groups, improving the model's fairness and inclusivity. These advancements would contribute to developing AES systems that are not only accurate but also equitable and versatile across global educational settings.

## CONCLUSIONS

In this study, we introduced a novel approach to AES by combining SBERT embeddings with LSTM networks and attention mechanisms. Our results show that this hybrid architecture significantly improves prediction accuracy and reduces errors compared to traditional and state-of-the-art models. By leveraging SBERT's rich semantic representations and enhancing sequential processing through LSTM and attention, the model demonstrated superior performance across multiple metrics, including MSE, $R^2$, RMSE, MAE, and QWK.

Despite the model's success, limitations related to dataset diversity and computational demands remain. Future research should explore the application of more diverse datasets, advanced attention mechanisms, and optimization strategies to further refine the model's

scalability and adaptability. Overall, our study highlights the potential of integrating deep learning techniques in AES to provide more accurate, efficient, and reliable assessments, contributing to the ongoing advancement of educational technology.

### Funding
The authors received no funding for this work.

### Competing Interests
The authors declare that they have no competing interests.

### Author Contributions
- Yuzhe Nie conceived and designed the experiments, performed the experiments, analyzed the data, performed the computation work, prepared figures and/or tables, authored or reviewed drafts of the article, and approved the final draft.

### Data Availability
The code and data used in the experiments are available in the Supplemental Files.

The data is originally from the Kaggle competition The Hewlett Foundation: Automated Essay Scoring: https://www.kaggle.com/competitions/asap-aes.

### Supplemental Information
Supplemental information for this article can be found online at http://dx.doi.org/10.7717/peerj-cs.2634#supplemental-information.

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
