# Peer review of "Automated essay scoring with SBERT embeddings and LSTM-Attention networks"

_PeerJ Computer Science, doi:10.7717/peerj-cs.2634_

## Round 0.1 · original submission · Major Revisions

The study requires major revisions based on comments from both reviewers. Specifically, they recommend conducting additional experiments, such as comparing the proposed method with a gradient-boosting algorithm and the approach proposed by Wang et al. (2022), which is already mentioned in the manuscript.

Reviewer 1 ·

Basic reporting

+ The manuscript is well-written, using clear and technical language. The introduction and background sections give a solid foundation for the study; nevertheless, they may benefit from including more relevant research. The figures are of good quality, given in vector format, and seamlessly incorporated into the manuscript. They are clearly identified, detailed, and provide excellent support for the material.
+ The code and data are published as supplemental documents, making them freely available and reproducible, hence increasing the study's transparency and usability.

Experimental design

+ The methods are described with sufficient detail and information to allow replication, as the code and data are provided. The discussion on data preprocessing is adequate. The evaluation methods, assessment metrics, and model selection techniques are also well-detailed.
+ Although all sources are generally well-cited, it appears that some references are missing for lines 95–96 on page 3 (Vaswani et al., 2017; Huang et al., 2020; Badaro et al., 2023; Mao, 2024).

Validity of the findings

+ The experiments and evaluations are generally well-executed and satisfactory. However, it is notable that while the authors tested several machine learning models alongside BERT (as shown in Table 2), gradient-boosting algorithms, such as XGBoost, LightGBM, or CatBoost, were not included. These algorithms are well-regarded for their performance in many tasks and could provide valuable comparative insights. Including at least one gradient-boosting model in future experiments would strengthen the evaluation.
+ Additionally, while the authors discussed the limitations of the proposed method, the manuscript would benefit from a clearer outline of potential directions for future work. Suggestions for improvements or extensions could enhance the study's impact and guide further research in this area.

Additional comments

Please double-check the capitalization of some items in the reference list. For example, "bert" should be written as "BERT".

Cite this review as

Reviewer 2 ·

Basic reporting

The manuscript is well-structured and written in clear, formal English. The literature review is comprehensive and provides a thorough overview of the topic. The tables and figures are well-designed and effectively present the research findings. To further improve the manuscript, consider including additional visualizations and addressing the specific points for clarification outlined below.

Experimental design

- The proposed method is described in sufficient detail, and the provision of code and data ensures that the reported results can be reproduced.
- Key aspects such as data preprocessing, evaluation methods, assessment metrics, and model selection methods are all adequately explained.
- While the authors compared their method with other machine learning and deep learning approaches, the comparison could be expanded. For instance, although the study by Wang et al. (2022), mentioned in the references, is relevant, it was not included in the comparison. If possible, please incorporate a comparison with Wang et al.'s method.
- Could you elaborate on the rationale behind selecting the two loss functions used in the paper?

Validity of the findings

- While the experiments and evaluations were performed satisfactorily, it would be better if the authors to repeat the experiments multiple times to achieve a more reliable estimation of performance.
- Also, incorporating more visualizations, such as those illustrating the predictions, would enhance the presentation and understanding of the results.

Additional comments

Currently, the names of all conferences in the list of references are in lowercase. Please update them to the correct formatting.

Cite this review as

---

## Round 0.2 · accepted · Accept

The authors have thoroughly addressed all of the reviewers' comments, and in alignment with the reviewers' suggestions, the manuscript is now ready for publication.

Reviewer 1 ·

Basic reporting

No comment. The revisions look good.

Experimental design

No comment.

Validity of the findings

The authors have incorporated additional models, and the updated experimental results further support the effectiveness of the proposed method. This revision also outlines future directions for the study.

Additional comments

No further comments.

Cite this review as

Reviewer 2 ·

Basic reporting

The revised version is well-organized and clearly written, with a thorough literature review. I don't have any other comments.

Experimental design

The authors have conducted additional experiments and provided a clear rationale for selecting the two loss functions used in the paper. I have no further comments.

Validity of the findings

The authors have repeated the experiments multiple times to obtain a more reliable performance estimation. I have no further comments.

Additional comments

No further comments.

Cite this review as